# Detection of Lymphocytic Choriomeningitis Virus in House Mouse (*Mus musculus*) in Brazil

**DOI:** 10.3390/v17121544

**Published:** 2025-11-26

**Authors:** Gabriel Rosa Cavalcanti, Jorlan Fernandes, Fernando de Oliveira Santos, Bernardo Rodrigues Teixeira, Alexandro Guterres, Julia Brignone, Silvana Levis, Camila dos Santos Lucio, Sócrates Fraga da Costa-Neto, Vagner Fonseca, Marta Giovanetti, Luiz Carlos Junior Alcantara, Paulo Sérgio D’Andrea, Elba Regina Sampaio de Lemos, Renata Carvalho de Oliveira

**Affiliations:** 1Laboratory of Hantaviruses and Rickettsiosis, Oswaldo Cruz Foundation, Oswaldo Cruz Institute, Rio de Janeiro 21040-360, Rio de Janeiro, Brazil; gabrielrosa.97@gmail.com (G.R.C.); jorlan@ioc.fiocruz.br (J.F.); elemos@ioc.fiocruz.br (E.R.S.d.L.); 2Laboratory of Biology and Parasitology of Wild Reservoir Mammals (LABPMR), Oswaldo Cruz Foundation, Oswaldo Cruz Institute, Rio de Janeiro 21040-360, Rio de Janeiro, Brazil; fernando.oliveira.snts@gmail.com (F.d.O.S.); brt@ioc.fiocruz.br (B.R.T.); camila.lucio@ioc.fiocruz.br (C.d.S.L.); dandrea@ioc.fiocruz.br (P.S.D.); 3Department of Genetics, Ribeirão Preto Medical School, University of São Paulo—USP, Ribeirão Preto 14049-900, São Paulo, Brazil; guterres@fmrp.usp.br; 4Instituto Nacional de Enfermedades Virales Humanas Dr. Julio I. Maiztegui, INEVH-ANLIS, Monteagudo 2510, Pergamino 2700, Argentina; jubrignone@yahoo.com.ar (J.B.); slevis0@yahoo.com (S.L.); 5Fiocruz Mata Atlântica, Oswaldo Cruz Fundation, Rio de Janeiro 22713-560, Rio de Janeiro, Brazil; socrates.neto@fiocruz.br; 6Department of Exact and Earth Sciences, University of the State of Bahia, Salvador 41150-000, Bahia, Brazil; vagner.fonseca@gmail.com; 7Centre for Epidemic Response and Innovation, School of Data Science and Computational Thinking, Stellenbosch University, Stellenbosch 7600, South Africa; 8Instituto Rene Rachou, Fundação Oswaldo Cruz, Belo Horizonte 30190-002, Minas Gerais, Brazil; giovanetti.marta@gmail.com (M.G.); alcantaraluiz42@gmail.com (L.C.J.A.); 9Department of Sciences and Technologies for Sustainable Development and One Health, Università Campus Bio-Medico di Roma, 00128 Rome, Italy

**Keywords:** *Lymphocytic choriomeningitis virus*, *Mus musculus*, Brazil, house mouse, *Rattus*

## Abstract

The lymphocytic choriomeningitis virus (LCMV) is an under-investigated rodent-borne arenavirus primarily associated with its natural reservoir, the cosmopolitan rodent *Mus musculus*. Although widely distributed in mice worldwide, human cases are rare, likely under-reported, and often misdiagnosed. While typically asymptomatic or self-limiting, infection can progress to neurological disease, severe congenital outcomes, or fatal illness in transplant recipients. Despite its public health relevance, this study provides the first detection and characterization of LCMV in Brazil. We analyzed 236 rodent serum samples and 78 tissue samples from synanthropic rodents (*Mus musculus*, *Rattus rattus*, and *Rattus norvegicus*) collected during seven independent expeditions across the state of Rio de Janeiro, Southeastern Brazil. Using ELISAs, IgG anti-LCMV antibodies were detected in 20% of rodents, including two *R. rattus* (2/10), two *R. norvegicus* (2/95), and forty-five *M. musculus* (45/131). The LCMV’s RNA was amplified and partially sequenced from fourteen *M. musculus*, and complete S segment sequences were obtained from two mouse samples. Phylogenetic analyses revealed that these Brazilian strains belong to lineage I, which is composed of strains that induce disease in humans. Our results underscore the importance of implementing integrated surveillance for this zoonosis in Brazil.

## 1. Introduction

The lymphocytic choriomeningitis virus (LCMV—*Mammarenavirus choriomeningitidis*) is a neglected rodent-borne virus and a prototype of the *Arenaviridae* family distributed worldwide. LCMV is an enveloped single-stranded negative-sense RNA virus with an average size of 110 to 130 nm in diameter and two ambisense RNA segments, small (S) and large (L), which transcribe the nucleoprotein (NP) and the glycoprotein precursor (GPC); zinc-binding matrix protein (Z); and RNA-dependent RNA polymerase (RdRp) [1,2,3]. The primary reservoir of the LCMV is the cosmopolitan house mouse (*Mus musculus*) [4,5], but it has already been detected in wild rodents, in specimens of the *Rattus* genus [6], and in pet mice and hamsters [7]. Despite the widespread prevalence of LCMV in mice across Africa, Europe, Asia, and the Americas, human cases remain rarely documented and are likely under-reported and misdiagnosed, which hinders accurate estimates of incidence and prevalence rates [8].

Transmission in humans occurs mainly through the inhalation of aerosols associated with the excreta of infected rodents, but it can also occur vertically during pregnancy (transplacentally) [5]. An LCMV infection in immunocompetent individuals is usually asymptomatic and self-limiting; however, in some cases, it may progress to meningitis and meningoencephalitis [9]. Prenatal transmission is associated with severe outcomes, including congenital malformations such as microcephaly and miscarriage, with an estimated mortality rate around 35% [10]. In immunocompromised people, several clusters of fatal donor-derived LCMV infections of solid organ transplant recipients were reported, resulting, in most cases, in multisystem organ failure and a very high fatality rate [11].

Mammarenaviruses have been documented in Brazil since the 1960s, with the identification of New World arenaviruses in rodents from distinct clades [12,13,14], including the Sabia virus (*Mammarenavirus brazilense*) in the 1990s [15]. Despite reports in neighboring South American countries, such as Argentina, Colombia, and French Guiana, knowledge about LCMV in Brazil has remained limited, relying primarily on serological evidence from a 2008 serosurvey in which antibodies were detected in 2% of animal-handling professionals [16,17,18,19,20]. These findings in South America, combined with the worldwide distribution of the LCMV’s main reservoirs, motivated us to investigate its presence in synanthropic rodents in different localities of the state of Rio de Janeiro, a coastal region of Brazil characterized by port areas and precarious socioeconomic and structural conditions. These conditions favor the presence and high infestation rates of commensal rodents, such as *Mus* and *Rattus* species, with infestation rates in some Brazilian slums reaching up to 45.9%, creating environments conducive to the circulation and spread of this rodent-borne virus [21,22,23].

This study provides the first detection and characterization of LCMV in Brazil, circulating in its primary reservoir, the common house mouse, *M. musculus*. This represents an important alert for Brazilian public health authorities and underscores the need to implement and strengthen active surveillance programs for this zoonotic disease, which is currently unknown in the country.

## 2. Materials and Methods

### 2.1. Study Areas and Rodent Sampling

Synanthropic rodents were recorded at seven locations across six municipalities in the state of Rio de Janeiro, Southeastern Brazil, from 2008 to 2018. The state predominately consists of Atlantic Forest fragments, pasture, agriculture and rural areas, coastal sand dune vegetation, and highly populated and urbanized areas [24]. The study areas were in the municipalities of Rio de Janeiro (urbanized area), Campos dos Goytacazes (sugarcane plantation), São Gonçalo (urbanized area), Valença (edge of forest fragments within an agricultural matrix), Silva Jardim, and Casimiro de Abreu. The last two areas were grassy areas between forest fragments located in conservation units, namely the Poço das Antas Biological Reserve (“Ilha dos Barbados”) and the São João River Basin/Golden Lion Tamarin Environmental Protection Area (APA), respectively. These locations were selected to encompass a diverse range of environments—including urban, peri-urban, rural, and open-area matrices—in order to capture ecological variability that could influence virus circulation. Because there were no previous reports of infection in rodents and the geographic distribution of the disease within the country remained undefined, this broad sampling strategy was adopted to maximize the likelihood of virus detection across different habitats and in different populations of *Mus* and *Rattus*.

In each rodent trapping locality, transects consisting of 15 to 20 capture stations spaced 20 m apart were established. Each station contained two live traps—one Tomahawk^®^ (Model 201, 16 in × 5 in × 5 in, Hazelhurst, WI, USA) and one Sherman^®^ (Model XLK, 3 in × 3.75 in × 12 in, Tallahassee, FL, USA)—suitable for capturing small mammals weighing up to 3 kg. The transects were placed 300 m apart, following the methodology of Püttker et al. [25]. Sherman and Tomahawk traps were baited with a mixture of peanut butter, banana, oats, and bacon and were checked every morning for five consecutive days.

Each animal captured was taken to a field laboratory, anesthetized, euthanized, and necropsied for the collection of biological samples (blood and kidney), and biometric, bionic, and reproductive data were recorded [26]. Taxonomic identification of the species was based on cranial and external body morphology and confirmed by karyotypic analysis, using cytogenetic techniques, when necessary [27,28]. The procedures for capturing the animals followed the methods of the biosecurity manuals for handling wild animals adopted by the Center of Disease Control (CDC, Atlanta, GA, USA) and Fundação Oswaldo Cruz (FIOCRUZ, Rio de Janeiro, Brazil) [26,29]. The fieldwork and the collection of zoological material were authorized by the Brazilian Ministry of the Environment (permanent license No. 13373/2007 and authorization No. 58374/2017, MMA/ICMBio/SISBIO) and by the FIOCRUZ Animal Ethics Committee through licenses CEUA L-049/2008, LW-39/2014, and L-036/2018.

### 2.2. Serological Analysis

Rodent’s sera were evaluated with an IgG enzyme-linked immunosorbent assay (ELISA) following the protocol of the Instituto Nacional de Enfermedades Virales Humanas “Julio I. Maiztegui” using inactivated antigens obtained from Vero E6 cell cultures [30].

Briefly, 96-round-bottom-well microplates (Thermo Scientific^TM^, Waltham, MA, USA) were coated with 100 µL of the cell lysate diluted in phosphate-buffered saline (PBS) with a pH of 7.4. One half of the plate contained the infected cell lysate (LCMV), and the other half contained the uninfected cell lysate (Vero C76-ATCC^®^ CRL-1587™, Manassas, VA, US). The plates were kept overnight at 4 °C and then washed five times with 0.1% Tween 20 (Merck & Co., Inc., Kenilworth, NJ, USA) in PBS (PBS-T). The wells were then filled with 100 µL of diluted test sera at 1:400 dilution. As diluent, PBS with 0.1% Tween 20 (Merck & Co., Inc., Kenilworth, NJ, USA) and 5.0% skim milk (BD Difco™, Franklin Lakes, NJ, USA) was used. The plates were incubated for 1 h at 37 °C. After five washes, 100 µL of goat Anti-Mouse IgG and Anti-Rat IgG linked to peroxidase (*v*/*v*) (Kirkegaard & Perry Laboratories, KPL, Gaithersburg, MD, USA) at 1:2000 dilution was placed in each well, and the plates were incubated for 1 h at 37 °C. The five washes were repeated, and 100 µL of ABTS [2, 2′-azino-di (3-ethylbenzthiazoline-6-sulfonate)] substrate (KPL, Gaithersburg, MD, USA) was added to each well. The plates were kept for 30 min at 37 °C. Absorbance reading was performed at a wavelength between 405 nm and 450 nm using a spectrophotometer (Microplate Reader, LMR-96, Kasuaki). The optical densities (ODs) obtained from the sera reacting to the LCMV antigen were subtracted from the OD values obtained from the same sera reacting to the non-specific antigen. The sample’s reactivity parameters were titer level ≥1/400 and final DO above the cut-off ≥0.2.

### 2.3. RT-PCR and Phylogenetic Analysis

Rodent kidney samples were subjected to viral RNA extraction using the PureLink™ Micro-to-Midi Total RNA Purification Kit (Invitrogen, San Diego, CA, USA). Subsequently, 2 µL of the extracted RNA was used for cDNA synthesis and amplification with the SuperScript™ III One-Step RT-PCR System with Platinum^®^ Taq DNA Polymerase (Invitrogen, San Diego, CA, USA). For the second PCR (PCR 2), the AmpliTaq Gold™ DNA Polymerase with Buffer II and MgCl_2_ Kit (Invitrogen, San Diego, CA, USA) was employed.

The first PCR (PCR 1) was performed using oligonucleotide primers that amplify a 660 bp fragment of the viral nucleoprotein coding region: 1817V-LCM (5′-AIA TGA TGC AGT CCA TGA GTG CAC A-3′) and 2477C-LCM (5′-TCA GGT GAA GGR TGG CCA TAC AT-3′). A nested PCR was then performed using 2 µL of the PCR 1 product with the primers 1902V-LCM (5′-CCA GCC ATA TTT GTC CCA CAC TTT-3′) and 2346C-LCM (5′-AGC AGC AGG YCC RCC TCA GGT-3′) [31]. PCRs were performed in a Veriti™ 96-Well Fast Thermal Cycler (Applied Biosystems™, Foster City, CA, USA). PCR 1 conditions were as follows: 48 °C for 45 min; 94 °C for 2 min; 40 cycles of 94 °C for 30 s, 61 °C for 40 s, and 68 °C for 50 s; followed by a final extension at 68 °C for 5 min. PCR 2 conditions were as follows: 94 °C for 5 min; 35 cycles of 94 °C for 30 s, 57 °C for 30 s, and 72 °C for 50 s; and a final extension at 72 °C for 7 min.

Positive amplicons were purified from 1.5% agarose gels using the Wizard^®^ Genomic DNA Purification Kit (Promega, Madison, WI, USA) and subjected to nucleotide sequencing with the BigDye^®^ Terminator v3.1 Cycle Sequencing Kit (Applied Biosystems, Foster City, CA, USA), following the manufacturer’s instructions. Sequencing reactions were run on an ABI Prism 3130x Genetic Analyzer (Applied Biosystems).

In parallel, sequencing was also performed on the Oxford Nanopore Technologies (ONT) MinION platform using the SMART-9N approach, as previously described by Claro et al. [32]. This random-primed, sequencing agnostic method generates double-tagged cDNA through template switching, followed by PCR amplification and library preparation for long-read nanopore sequencing. This approach enables sensitive detection from low RNA input and provides broad viral genome coverage without the need for virus-specific primers. Raw nanopore reads were processed and assembled using Genome Detective [33], which facilitated automated classification and consensus genome reconstruction.

Phylogenetic analyses were conducted using both newly generated sequences from this study and reference sequences retrieved from GenBank. Multiple sequence alignments were performed with MAFFT v7.526 using the E-INS-i algorithm, which is suitable for sequences with multiple conserved domains and long gaps. Maximum likelihood (ML) analyses were carried out in IQ-TREE v3.0.1, with the best-fit substitution model selected by ModelFinder under the Bayesian Information Criterion (BIC), which identified TVM+F+I+G4 as the optimal model. The ML tree was inferred with 1000 ultrafast bootstrap replicates (-B 1000) and 1000 SH-aLRT replicates (-alrt 1000), with four computational threads (-T 4) to ensure robust branch support. Additionally, Bayesian phylogenetic inference was performed in MrBayes v3.27a, using the GTR substitution model with gamma-distributed rate variation across sites and a proportion of invariable sites (GTR + I + Γ). Two independent Markov Chain Monte Carlo (MCMC) runs with four chains each were executed for 3,000,000 generations, sampling every 100th generation, resulting in 30,000 trees. After discarding the initial 10% as burn-in, a majority-rule consensus tree was generated. The analysis converged successfully, as indicated by the average standard deviation of split frequencies falling below 0.05 [34,35,36,37,38].

A matrix of genetic distances was produced and estimated using the MEGA 12 software following the Pairwise Distance method with 1000 bootstrap replications to assess variance; the p-distance model included both transitions and transversions, under uniform rates among sites and a homogeneous pattern among lineages, analyzing all codons. The protein analysis used the same bootstrap parameters [39].

## 3. Results

### 3.1. Serological Results

A total of 236 samples of synanthropic rodents, including *R. rattus* (6), *R. norvegicus* (99), and *M. musculus* (131), were analyzed from all expeditions. Anti-LCMV IgG antibodies were detected in 49 rodent samples, showing an overall seroprevalence of 20%, from the municipalities of Campos dos Goytacazes, São Gonçalo and Silva Jardim, Rio de Janeiro state, including 20 females and 25 males of the *M. musculus* species (n = 45), 1 male and 1 female of the *R. rattus* species (n = 02), and 2 males of the *R. norvegicus* species (n = 02) (Table 1 and Figure 1).

### 3.2. Virus Characterization

A total of 78 kidney samples were available for RT-PCR analysis. The positive animals, 18% (14/78), belonged to the species *M. musculus* (10 males and 4 females), which were all collected in sugarcane plantations of the municipality of Campos dos Goytacazes (Figure 1, Table 1). Only two tissue samples were obtained from Silva Jardim, seven were collected from Casimiro de Abreu, and none were positive. Of the 41 seroreactive samples analyzed, only 1 tested positive in the RT-PCR, and it was collected in Campos dos Goytacazes.

The comparison of the nucleotide (nt) and amino acid (aa) nucleoprotein sequences with other LCMV strains available in GenBank confirmed the identification of LCMV in all RT-PCR-positive rodents (Appendix A). A comparative analysis of the complete viral S segment sequences from LBCE20151 (GenBank accession no. PX445789) and LBCE20173 (GenBank accession no. PX445790) revealed inter-strain variability, with an 86% and 87% nucleotide similarity in the GPC and NP genes, respectively, and a 97% amino acid identity for both proteins. When compared to other known LCMV strains, the viral Brazilian strain LBCE20151 showed the highest nucleotide (87%) and amino acid (98%) identity in the complete NP gene, as well as an 85% nt and 95% aa identity in the GPC gene, with the Douglas-4707 LCMV strain (GenBank accession no. FJ607035) identified from a human CSF sample in the USA. The Brazilian viral strain LBCE20173 demonstrated the highest nt (86%) and aa (97%) identities with four strains identified in the USA in the NP gene: Douglas-4707 (GenBank FJ607035); IN-2012 (KF732824); 810,885 (FJ607032); and Traub (DQ868487). In the GPC gene, the highest nt (85%) was observed with different USA strains and in aa (96%) with CLONE13 (DQ361065), also from the USA [40] (Table 2).

The phylogenetic analysis based on complete S segment sequences showed that the Brazilian strains LBCE20151 and LBCE20173 are most closely related to the Makokou strain (GenBank accession no. KM523323), isolated from a house mouse (*Mus musculus domesticus*) in Gabon in 2012 [41], forming a distinct subclade within lineage I (Figure 2). The S tree’s topology is congruent with another tree generated in a recent study [42]. The complete L segment sequence could not be obtained from the two samples analyzed with the MinION platform due to the low sequencing coverage.

## 4. Discussion

The LCMV is likely circulating worldwide due to the cosmopolitan distribution of its reservoirs. Epidemiological studies have reported LCMV antibodies in a significant proportion of wild mice, *M. musculus* populations, mainly in urban areas, as demonstrated in Argentina (6.9–20.1%), and in the municipality of Sucre in Colombia (10%) [17,43]. In the present study, we found a notably high seroprevalence of LCMV among mice (34%), with 21% of *M. musculus* captured in Campos dos Goytacazes and 19% of those captured in Silva Jardim testing as seropositive.

The localities where rodents were captured were predominantly characterized by open landscapes. In Silva Jardim, these consisted of grassy areas between forest fragments, while in Campos dos Goytacazes (the fifth largest urbanized area in the state [24]), the environment was composed mainly of sugarcane fields interspersed with rural and urbanized areas. In Campos, an investigation of the diet of Barn Owls (*Tyto furcata*) revealed a high abundance of *M. musculus*, contrasting with the low representation of native rodents. This finding reflects reduced species richness and the dominance of house mice in urban environments [44]. In addition to the urban matrix, the presence of the generalist *M. musculus* in more open areas and forest fragments has also been reported, with evidence of its invasion and integration into these habitats through the sharing of helminth species with wild rodents such as *Akodon cursor*, which is uncommon for mice [45]. A previous study reported a higher abundance of small rodents, particularly *M. musculus*, in sugarcane plantations compared to other types of landscapes [46]. This geographic and ecological context may explain the high number of *M. musculus* captured in the region.

Furthermore, a study conducted in Italy demonstrated a positive correlation between the rodent population density and LCMV antibody prevalence, supporting the high seroprevalence observed in our study [47]. Similarly, our findings in mice align with results from the Istrian Peninsula (Northern Adriatic, shared by Italy, Slovenia, and Croatia), where Duh et al. (2017) reported a 47.37% seroprevalence of LCMV in the commensal *M. musculus* inhabiting areas characterized by a high anthropogenic impact, extensive waste accumulation, and a close proximity to human settlements [48].

Although house mice are more likely to be infected than other rodent species, LCMV has been reported in a range of wild rodents. For example, Blasdell et al. (2008) reported its occurrence in the United Kingdom, including in Norway rats (*R. norvegicus*) and *Apodemus* sp., with seroprevalence varying between species (1.4–26%) and across trapping sites, ranging from 0.0 to 50.0% [6]. We also detected LCMV antibodies in *Rattus*—two *R. norvegicus* and two *R. rattus*—from the Campos dos Goytacazes and São Gonçalo municipalities, demonstrating the potential for spillover into other sympatric species that may be involved in the enzootic transmission cycle and in the dissemination of LCMV in Rio de Janeiro. The close contact between *M. musculus* and other native rodents, including wild Sigmodontinae species from the American continent, could facilitate the interspecific transmission of LCMV, thereby increasing the risk of human infection.

In our study, all LCMV RNA-positive samples were obtained from *M. musculus* specimens (14) collected in the Campos dos Goytacazes municipality. Genetic and phylogenetic analyses have revealed that LCMV exhibits high genetic diversity among strains from both the New and Old World, comprising five major lineages (I–V). Notably, LCMV lineages show no clear geographic clustering, reflecting the complex phylogeographic history of the house mouse and its spread through human-mediated transport and trade [40,49,50]. Our phylogenetic analyses of the complete LCMV S RNA sequence confirmed that the Brazilian LCMV strains on S complete sequences (LBCE20151 and LBCE20173) are most closely related to the Makokou strain from Gabon, belonging to linage I [41]. Globally, most LCMV strains cluster within lineage I, which includes strains that are associated with severe disease in humans.

According to our genetic analysis, the LBCE20151 and LBCE20173 strains showed the highest nucleotide and amino acid identities with several North American isolates. These findings indicate a close genetic relationship between Brazilian, North American, and Gabonese strains, suggesting potential historical introductions or shared ancestral lineages. This apparent connection reinforces the notion that LCMV lacks strict geographical clustering and that potential routes of viral introduction warrant further investigation. Together, these findings expand the known geographic range of lineage I and underscore the importance of continuous molecular surveillance to clarify the evolutionary dynamics and potential epidemiological implications of the LCMV’s circulation in Brazil.

We also compared the GPC and NP gene sequences of the Brazilian strains LBCE20151 and LBCE20173 to LCMV strains with well-characterized mutations associated with changes in viral replication and infection outcomes. It has been demonstrated that amino acid substitutions in the GP1 subunit, such as residue 260 (GP1 F260L) in the LCMV Clone 13 strain, can modulate infection and suppress the CTL response, leading to persistent infection in mice, whereas the Armstrong strain induces acute infection [51,52]. The GP1 F260L substitution alters the virus’s receptor-binding properties, enhancing its interaction with the host α-dystroglycan receptor and promoting the increased infection of dendritic cells. This, in turn, modifies the host’s immune response and contributes to viral persistence [52,53]. Another amino acid change in GP1—N176D, observed in at least half of the persistent LCMV strains—was also detected in both of our sequences, although it does not appear to directly contribute to the persistence mechanism [52]. Another viral strategy to evade the host’s immune system involves the anti-type I interferon response, which targets the inhibition of interferon regulatory factor 3 (IRF3) activation. In this mechanism, several amino acid residues of the NP protein play a critical role, including the NP DIEGR motif (D382A, G385A, and R386A) and the NP 3′–5′ exonuclease motif (D382A, E384A, D459A, H517A, and D552A) [54,55,56]. These regions were identified in the Brazilian sequences (LBCE20151 and LBCE20173). These conserved motifs highlight the functional importance of NP-mediated immune evasion in LCMV and underscore the need for further studies exploring how these mechanisms contribute to viral persistence and pathogenesis in natural hosts. Additionally, these findings suggest that the Brazilian LCMV strains share conserved mechanisms of immune evasion with other lineages worldwide, reflecting the evolutionary stability of these functional domains.

The proximity of our sequence to LCMV strains known to infect humans raises concerns about potential human infections in the study areas, particularly in regions with high rodent infestation rates and frequent human contact with these commensal rodents. In this context, it is also important to consider the possibility of cases being misdiagnosed as other endemic diseases and therefore not being reported, due to the lack of awareness of LCMV in Brazil. Moreover, since most LCMV infections are asymptomatic or present as a mild influenza-like illness, the true incidence and prevalence of LCMV in the country remain unknown [20,57]. Globally, epidemiological studies indicate that the highest LCMV seropositivity has been observed among individuals with frequent contact with rodents, in both urban and rural settings, including seroprevalence rates of up to 47% among the employees of rodent breeding facilities during the 2012 outbreak [58] and 37% in a rural population on a small island in Croatia [59].

Our study provides the first evidence of LCMV circulation in Brazil. Although the sampling was geographically limited to the state of Rio de Janeiro, the results establish a foundation for future investigations across other Brazilian regions with diverse ecological and socioeconomic conditions. The restricted number of rodent tissue samples available for the RT-PCR analysis—particularly from areas outside of Campos dos Goytacazes—limited the assessment of the broader geographic distribution of active infection. Nevertheless, serological evidence supports viral circulation in other municipalities and indicates a potential dispersion of three synanthropic rodent species across the state of Rio de Janeiro: *M. musculus*, *R. norvegicus*, and *R. rattus*. These findings underscore the importance of continued molecular surveillance and genomic studies to improve our understanding of the epidemiology, diversity, and evolutionary dynamics of LCMV throughout the country.

Considering the risk of human infection and the identification of LCMV in rodents, particularly in M. musculus, in the state of Rio de Janeiro, the need for the integrated surveillance of this zoonosis in Brazil is apparent. Such efforts should encompass the systematic monitoring of rodent populations, especially in areas with high levels of mouse and rat infestations; seroprevalence studies in at-risk human populations; and the inclusion of LCMV in the differential diagnosis of other endemic diseases, such as viral meningitis in immunocompetent individuals and congenital syndromes associated with TORCH and Zika virus infections in pregnant women. In this way, it will be possible to better understand the true extent of LCMV’s circulation and its impact on public health.

## Figures and Tables

**Figure 1 viruses-17-01544-f001:**
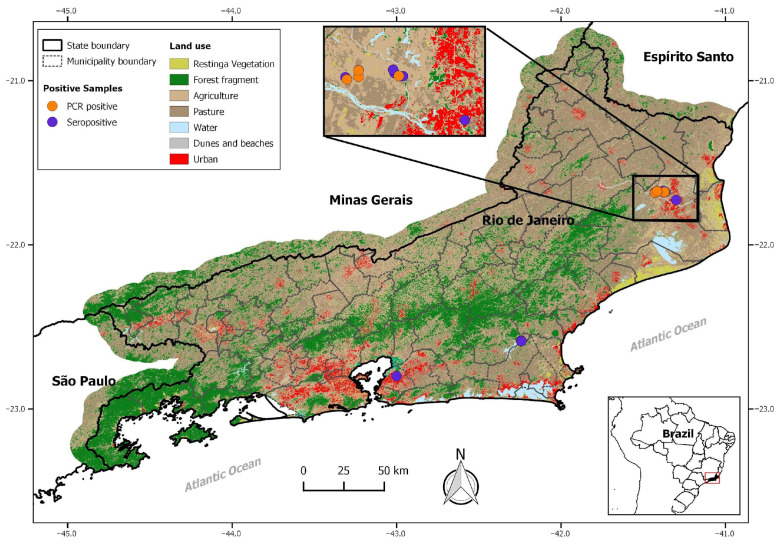
A map of the state of Rio de Janeiro showing the locations where seroreactive (purple dots) and RT-PCR-positive (orange dots) samples were detected. The inset highlights the northern region of the state, corresponding to the municipality of Campos dos Goytacazes, where both serological and molecular detections were recorded. The map also includes general information on land use, vegetation cover, and urbanization, illustrating the environmental diversity of the sampled areas.

**Figure 2 viruses-17-01544-f002:**
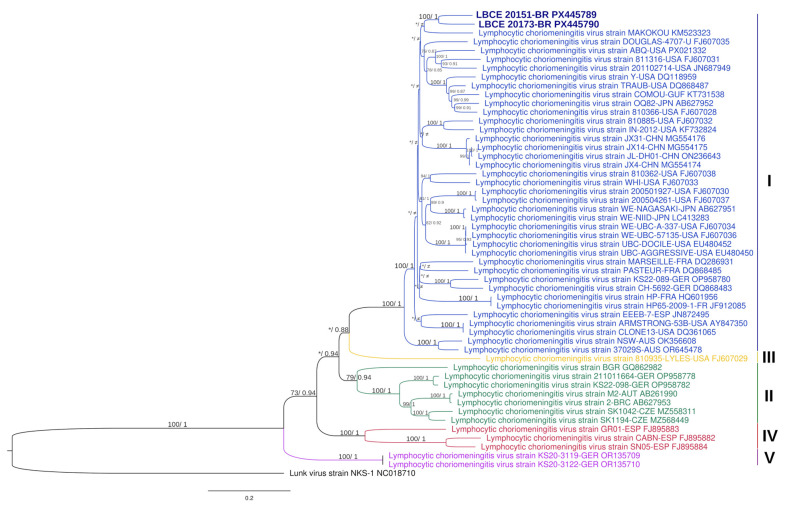
Phylogenetic trees based on the S segment sequences of the LCMV estimated using maximum likelihood (ML) and Bayesian inference. The ML tree is shown here, rooted with the outgroup sequence of the Lunk virus strain NKS-1 (GenBank: NC_018710). Numbers above branches indicate ML bootstrap support values (IQ-TREE) and Bayesian posterior probabilities (MrBayes), respectively, shown as (>70%/>0.7). Asterisks (*) indicate values below these thresholds, and the (≠) symbol denotes topological differences between ML and Bayesian trees. GenBank accession numbers are shown at the end of each strain name, and colors represent the five distinct LCMV lineages.

**Table 1 viruses-17-01544-t001:** Summary of the municipalities, species, and number of rodents tested by ELISA and RT-Nested PCR and the results.

Municipalities and Rodent Species		LCMV Antibodies	RT-Nested PCR
		Positive/N Total (%)	Positive/N Total (%)
Rio de Janeiro			
	*R. norvegicus*	0/4	NA
Campos dos Goytacazes			
	*M. musculus*	38/68 (55)	14/64 (21)
	*R. norvegicus*	1/2 (50)	0/1
	*R. rattus*	2/5 (40)	0/4
Silva Jardim			
	*M. musculus*	7/37 (19)	0/2
São Gonçalo			
	*M. musculus*	0/1	NA
	*R. norvegicus*	1/93 (1)	NA
Valença			
	*M. musculus*	0/17	NA
Casimiro de Abreu			
	*M. musculus*	0/8	0/6
	*R. rattus*	0/1	0/1
Total		49/236 (20)	14/78 (18)

**Table 2 viruses-17-01544-t002:** Amino acid and nucleotide similarities of LCMV LBCE strains compared with representative strains from each LCMV lineage (GPC and NP genes).

Strain_Country_Lineage	Accession Number	(%) Nucleotides and Amino Acids Similarities							
		LBCE20151				LBCE20173			
		NT		AA		NT		AA	
		GPC	NP	GPC	NP	GPC	NP	GPC	NP
COMOU_GUF_I	KT731538	86	85	95	96	83	84	93	95
MAKOKOU_GABN_I	KM523323	85	86	93	97	84	86	93	96
IN-2012_USA_I	KF732824	85	86	95	97	84	86	94	97
810885_USA_I	FJ607032	84	86	95	97	84	86	94	97
DOUGLAS_4707_USA_I	FJ607035	85	87	95	98	85	86	94	97
UBC_DOCILE_USA_I	EU480452	85	85	94	97	85	86	93	95
UBC_AGGRESSIVE_USA_I	EU480450	85	85	94	97	85	86	93	96
ARMSTRONG_53B_USA_I	AY847350	85	86	95	97	84	84	95	95
WE_UBC_57135_USA_I	FJ607036	85	85	94	97	85	86	93	96
JX31_CHN_I	MG554176	84	87	93	97	84	85	93	97
CLONE13_USA_I	DQ361065	85	86	95	97	84	84	96	95
TRAUB_USA_I	DQ868487	85	86	95	97	85	86	94	97
2_BRC_II	AB627953	78	81	90	93	78	80	89	93
CABN_ESP_IV	FJ895882	73	80	80	91	73	78	80	90
KS20-3122_GER_V	OR135710	75	79	85	93	75	78	84	92
810935_LYLES_USA_III	FJ607029	78	80	89	92	78	79	89	92

## Data Availability

The data used for comparison were obtained and are openly available for consultation in the NIH genetic sequence database (GenBank), and original data can be found under the following access numbers on Genbank: LBCE20151 (PX445789) and LBCE20173 (PX445790). All accession numbers of partial sequences are listed on Appendix A.

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
