# Peer review of "Detection of Lymphocytic Choriomeningitis Virus in House Mouse (Mus musculus) in Brazil"

_viruses, 2025, doi:10.3390/v17121544_

Round 1

Reviewer 1 Report

Comments and Suggestions for Authors

This paper studied the first detection and characterization of lymphocytic choriomeningitis virus (LCMV) in Brazil, which expand the known geographic region of LCMV transmission. This paper enables us to better understand the true extent of LCMV circulation and its potential epidemiological impact on global public health. However, there are several areas that require further revision to enhance the scientific rigor and clarity of the manuscript.

  • The Introduction provides a clear background of the research field and the transmission of LCMV infection. However, the supplement for the necessity and novelty of the present work could be strengthened.
  • The manuscript briefly mentions the Brazilian strains MusLBCE20151 and MusLBCE20173 are mostly related to the Makokou strain in Fig. 2. and Table 2. Considering the glycoprotein (GP) of LCMV serves as virus binding protein to its receptor on host cells and plays an important role in pathogenesis and epidemiology of the vir It is recommended that the authors should conduct a comprehensive GP sequence alignment of the representative strains. The Results should then be revised to incorporate the differences through linking identified key mutations to their potential functional consequences, which would greatly enrich the mechanistic insight of the study.
  • There are some questions that are not clearly clarified – what do we learn from these disparate results? What broad insight can we gain from the results? What is the significance of the findings? Therefore, the Discussion section would benefit from two key additions to enhance the scholarly rigor and perspective of the work. On one hand, a dedicated subsection or paragraph explicitly detailing the limitations of the work might be added. On the other hand, the conclusion of the Discussion should be expanded to provide a specific and actionable outlook for future research after this work.
  • The reasoning behind the selection of different locations is unclear. It is recommended that clearly statement should be added for better understanding.
  • Some references (e.g., Reference 3 and 4) include page numbers, while others (e.g., Reference 7 and 9) do not. Additionally, the format of Reference 41 is incorrect and needs to be verified and corrected. Please carefully check the references list under the journal's style guide and apply the correct format for all references.

Author Response

This paper studied the first detection and characterization of lymphocytic choriomeningitis virus (LCMV) in Brazil, which expand the known geographic region of LCMV transmission. This paper enables us to better understand the true extent of LCMV circulation and its potential epidemiological impact on global public health. However, there are several areas that require further revision to enhance the scientific rigor and clarity of the manuscript.

Response: We sincerely thank the reviewer for their insightful comments. All points raised have been thoroughly addressed in the revised manuscript:

The Introduction provides a clear background of the research field and the transmission of LCMV infection. However, the supplement for the necessity and novelty of the present work could be strengthened.

Response: We believe that the novelty of our findings in Brazil is noteworthy; however, as suggested, we have highlighted this information more clearly in the Introduction. The significance and public health impact of LCMV are highlighted in the opening paragraphs.

The manuscript briefly mentions the Brazilian strains MusLBCE20151 and MusLBCE20173 are mostly related to the Makokou strain in Fig. 2. and Table 2. Considering the glycoprotein (GP) of LCMV serves as virus binding protein to its receptor on host cells and plays an important role in pathogenesis and epidemiology of the vir It is recommended that the authors should conduct a comprehensive GP sequence alignment of the representative strains. The Results should then be revised to incorporate the differences through linking identified key mutations to their potential functional consequences, which would greatly enrich the mechanistic insight of the study.

Response: We thank the reviewer for this valuable suggestion. Additional information has been incorporated into the Discussion to address known mutations in LCMV glycoprotein gene and also NP gene and to discuss how the mutations identified in the Brazilian strains MusLBCE20151 and MusLBCE20173 relate to those previously reported.

There are some questions that are not clearly clarified – what do we learn from these disparate results? What broad insight can we gain from the results? What is the significance of the findings? Therefore, the Discussion section would benefit from two key additions to enhance the scholarly rigor and perspective of the work. On one hand, a dedicated subsection or paragraph explicitly detailing the limitations of the work might be added. On the other hand, the conclusion of the Discussion should be expanded to provide a specific and actionable outlook for future research after this work.

Response: Following your suggestion, we have included a paragraph in the Discussion highlighting the limitations of the study and emphasizing our new findings, which imply the need for further research to understand not only the prevalence and circulation of LCMV, but also the infection dynamics, including different rodent species, and evolutionary aspects.

The reasoning behind the selection of different locations is unclear. It is recommended that clearly statement should be added for better understanding.

Response: Yes, we have included additional information to clarify the rationale behind the selection of the study areas.

Some references (e.g., Reference 3 and 4) include page numbers, while others (e.g., Reference 7 and 9) do not. Additionally, the format of Reference 41 is incorrect and needs to be verified and corrected. Please carefully check the references list under the journal's style guide and apply the correct format for all references.

Response: The references have been double-checked and formatted according to the journal’s style guide.

Reviewer 2 Report

Comments and Suggestions for Authors

The authors investigate the prevalence of LCMV (serum IgG and genomic RNA) in rodents within the state of Rio de Janeiro, Brazil. Their findings add to a number of recent publications in the this area trying to generate more detailed data on LCMV prevalence and distribution. This is of significant interest considering that LCMV has caused lethal infection in primarily immunocompromised patients. In addition, LCMV serves as a good example of how global traffic has affected virus spread and evolution into different lineages.

Overall, the paper is of interest to a specialist audience; it is well written with only minor language issues (see below).

Major

  • All methods need to be described in sufficient detail without the need to refer back to published data. E.g. for serological analysis, the full detail should be given. Likewise, primer sequences and details of the PCR protocol are required.
  • Table 1 should more clearly indicate samples that were not suitable for PCR analysis as the result from these samples cannot be predicted and could significantly affect the interpretation of results.

Minor:

  • I don’t feel that the term “invisible” (abstract) is appropriate to describe a pathogen.
  • The authors mention authorisation but it is not clear if ethics approval was also given. This should be explicitly stated.
  • Some minor language issues are present, e.g. line 94/95. But overall the manuscript is easy to read and follow.
  • For RT-PCR, did they authors use random hexamer primers? Please confirm
  • I do appreciate the details for Figure 1, however, the blue dots are not easily seen.
  • Was there any overlap between PCR positive and Ig positive samples?
  • It could be due to the PDF but could the authors please check that the resolution for figure 2 is sufficient to read the details?
Comments on the Quality of English Language

see above comments. Some minor language issues are present. 

Author Response

Reviewer 2- The authors investigate the prevalence of LCMV (serum IgG and genomic RNA) in rodents within the state of Rio de Janeiro, Brazil. Their findings add to a number of recent publications in the this area trying to generate more detailed data on LCMV prevalence and distribution. This is of significant interest considering that LCMV has caused lethal infection in primarily immunocompromised patients. In addition, LCMV serves as a good example of how global traffic has affected virus spread and evolution into different lineages.

Overall, the paper is of interest to a specialist audience; it is well written with only minor language issues (see below).

Response: We sincerely thank the reviewer for their insightful comments. All points raised have been thoroughly addressed in the revised manuscript:

Major

All methods need to be described in sufficient detail without the need to refer back to published data. E.g. for serological analysis, the full detail should be given. Likewise, primer sequences and details of the PCR protocol are required.

Response: We have expanded the methodological section to include detailed descriptions of the serological analyses, primer sequences, and PCR protocols, as recommended.

Table 1 should more clearly indicate samples that were not suitable for PCR analysis as the result from these samples cannot be predicted and could significantly affect the interpretation of results.

Response: Yes, we have clarified this point in the Results section. A total of 78 kidney samples were available for RT-PCR analysis. Of the 41 seroreactive samples analyzed, only one tested positive by RT-PCR, and it was collected in Campos dos Goytacazes.

Minor:

I don’t feel that the term “invisible” (abstract) is appropriate to describe a pathogen.

Response: We agree with the comment and have removed the term from the abstract.

The authors mention authorisation but it is not clear if ethics approval was also given. This should be explicitly stated.

Response: We clarified the sentence to explicitly indicate that ethics approval was granted by the FIOCRUZ Animal Ethics Committee under licenses CEUA L-049/2008, LW-39/2014, and L-036/2018.

Some minor language issues are present, e.g. line 94/95. But overall the manuscript is easy to read and follow.

Response: Yes, we revised the sentence to improve clarity and flow in English. The manuscript has been reviewed for English language editing by MDPI (english-103363).

For RT-PCR, did they authors use random hexamer primers? Please confirm

Response: We included the following information in the text for clarification: “PCR 1 was performed using oligonucleotide primers that amplify a 660 bp fragment of the viral nucleoprotein coding region — 1817V-LCM (5′-AIA TGA TGC AGT CCA TGA GTG CAC A-3′) and 2477C-LCM (5′-TCA GGT GAA GGR TGG CCA TAC AT-3′). This was followed by a nested PCR using 2 µL of the PCR 1 product and the primers 1902V-LCM (5′-CCA GCC ATA TTT GTC CCA CAC TTT-3′) and 2346C-LCM (5′-AGC AGC AGG YCC RCC TCA GGT-3′), according to the method described by Emonet et al.(2007).”

I do appreciate the details for Figure 1, however, the blue dots are not easily seen.

Response: We have added more details regarding the map to improve clarity. The revised figure legend now specifies that the inset highlights the northern region of the state, corresponding to the municipality of Campos dos Goytacazes, where both serological and molecular detections were recorded.

Was there any overlap between PCR positive and Ig positive samples?

Response: We have included this information in the text. Of the 41 seroreactive samples analyzed, only one tested positive by RT-PCR, and it was collected in Campos dos Goytacazes.

It could be due to the PDF but could the authors please check that the resolution for figure 2 is sufficient to read the details?

Response: Yes, thank you. We have

Reviewer 3 Report

Comments and Suggestions for Authors

The manuscript by Cavalcanti et al. describes the first detection and characterization of LCMV in Brazil, revealing a 20% prevalence of IgG anti-LCMV antibodies in sampled rodents and the complete sequencing of the S segment from two mouse samples. As some of the identified LCMV sequences are closely related to those that have caused significant disease in the past, this study underscores the necessity for enhanced surveillance of this zoonotic disease in the region. The novelty of these findings is noteworthy, and they are recommended for publication.

Here are a few comments for consideration to improve clarity and impact –

1) It’s not clear how certain references are being used. For example, in line 100, reference 23 is listed but there is no indication what methods were implemented from that reference in the text. The same is true for line 108, reference 27, as it’s not clear if this is a CDC safety document or procedures from Fundação Oswaldo Cruz.

2) The S segments are referred to as MusLBCE20151 and MusLBCE20173 in the text, and LBCE20151_BR and LCBE20173_BR in table 2. There should be a consistent nomenclature.

3) The accession numbers should be added to the LBCE20151_BR and LCBE20173_BR strains in the phylogenetic tree in Figure 2.

4) Why was the L segment not sequenced? There should be a discussion regarding the attempts made or the issues encountered during the sequencing efforts if they were unsuccessful.

5) Additionally, a discussion could be included on how the new LCMV S segment sequences compare to known LCMV strains with well-characterized mutations associated with changes in viral replication and infection outcomes, as demonstrated in reference 40.

Comments on the Quality of English Language

Some of the sentences do not clearly translate to a more general audience. For example, in line 84, “07 localities in 06 municipalities of Rio de Janeiro state, Southeastern Brazil,” would be understood more clearly as “seven locations across six municipalities in the state of Rio de Janeiro, Southeastern Brazil.” Several examples, such as this one, exist in the manuscript.

Author Response

Reviewer 3 -The manuscript by Cavalcanti et al. describes the first detection and characterization of LCMV in Brazil, revealing a 20% prevalence of IgG anti-LCMV antibodies in sampled rodents and the complete sequencing of the S segment from two mouse samples. As some of the identified LCMV sequences are closely related to those that have caused significant disease in the past, this study underscores the necessity for enhanced surveillance of this zoonotic disease in the region. The novelty of these findings is noteworthy, and they are recommended for publication.

Here are a few comments for consideration to improve clarity and impact –

Response: We sincerely thank the reviewer for their insightful comments. All points raised have been thoroughly addressed in the revised manuscript:

1) It’s not clear how certain references are being used. For example, in line 100, reference 23 is listed but there is no indication what methods were implemented from that reference in the text. The same is true for line 108, reference 27, as it’s not clear if this is a CDC safety document or procedures from Fundação Oswaldo Cruz.

Response: We have indicated the references accordingly and included a new one, as pointed out in the methodology section.

2) The S segments are referred to as MusLBCE20151 and MusLBCE20173 in the text, and LBCE20151_BR and LCBE20173_BR in table 2. There should be a consistent nomenclature.

Response: Yes, we have standardized the nomenclature to LBCE20151 and LBCE20173 throughout the text.

3) The accession numbers should be added to the LBCE20151_BR and LCBE20173_BR strains in the phylogenetic tree in Figure 2.

Response: Yes, this has been done.

4) Why was the L segment not sequenced? There should be a discussion regarding the attempts made or the issues encountered during the sequencing efforts if they were unsuccessful.

Response: Yes, this information has been included in the Results section. The complete L segment sequence could not be obtained from the two samples analyzed with the MinION platform, most likely due to low sequencing coverage, which prevented full genome assembly.

5) Additionally, a discussion could be included on how the new LCMV S segment sequences compare to known LCMV strains with well-characterized mutations associated with changes in viral replication and infection outcomes, as demonstrated in reference 40.

Response: We have added a paragraph in the Discussion comparing the new LCMV S segment sequences obtained in this study with previously reported strains, focusing on well-characterized mutations known to influence viral replication and infection outcomes, as suggested.

Some of the sentences do not clearly translate to a more general audience. For example, in line 84, “07 localities in 06 municipalities of Rio de Janeiro state, Southeastern Brazil,” would be understood more clearly as “seven locations across six municipalities in the state of Rio de Janeiro, Southeastern Brazil.” Several examples, such as this one, exist in the manuscript.

Response:This sentence, as well as the entire document, has been reviewed for English language editing by MDPI (english-103363).

Round 2

Reviewer 1 Report

Comments and Suggestions for Authors

The authors’ revisions have significantly improved the manuscript, and most of my major concerns have been adequately addressed mainly by correcting mistakes, making their methods more transparent, improving the language and enhancing the scholarly rigor and perspective of the work. At this stage, I only have a few minor points that require the authors' attention before I can recommend the manuscript for publication:

  • Figure 2 Clarity: The current resolution of Figure 2 is somewhat low, making some details difficult to discern. Specifically, symbols like "≠" are blurry and hard to read. A higher-resolution figure would greatly improve readability.
  • Abbreviations Definition: On Abbreviations, the abbreviation “GPC” is defined as “Glycoprotein precursor”. For greater accuracy and clarity, this should be revised to “Glycoprotein precursor complex”.

Reviewer 2 Report

Comments and Suggestions for Authors

I thank the authors for incorporating my suggestions and recommendations.